# A common pathway for detergent-assisted oligomerization of Aβ42

Fidha Nazreen Kunnath Muhammedkutty[1], Ramesh Prasad [1], Yuan Gao[2], Tarunya Rao Sudarshan[2], Alicia S. Robang[2], Jens O. Watzlawik[3], Terrone L. Rosenberry[3], Anant K. Paravastu [2,4✉] & Huan-Xiang Zhou [1,5✉]

Amyloid beta (Aβ) aggregation is a slow process without seeding or assisted nucleation. Sodium dodecyl sulfate (SDS) micelles stabilize Aβ42 small oligomers (in the dimer to tetramer range); subsequent SDS removal leads to a 150-kD Aβ42 oligomer. Dodecylphosphorylcholine (DPC) micelles also stabilize an Aβ42 tetramer. Here we investigate the detergent-assisted oligomerization pathway by solid-state NMR spectroscopy and molecular dynamics simulations. SDS- and DPC-induced oligomers have the same structure, implying a common oligomerization pathway. An antiparallel β-sheet formed by the C-terminal region, the only stable structure in SDS and DPC micelles, is directly incorporated into the 150-kD oligomer. Three Gly residues (at positions 33, 37, and 38) create holes that are filled by the SDS and DPC hydrocarbon tails, thereby turning a potentially destabilizing feature into a stabilizing factor. These observations have implications for endogenous Aβ aggregation at cellular interfaces.

[1] Department of Chemistry, University of Illinois Chicago, Chicago, IL 60607, USA. [2] School of Chemical and Biomolecular Engineering, Georgia Institute of Technology, 311 Ferst Drive NW, Atlanta, GA 30332, USA. [3] Departments of Neuroscience and Pharmacology, Mayo Clinic, Jacksonville, FL 32224, USA. [4] Parker H. Petit Institute for Bioengineering and Biosciences, Georgia Institute of Technology, 315 Ferst Drive, Atlanta, GA 30332, USA. [5] Department of Physics, University of Illinois Chicago, Chicago, IL 60607, USA. ✉email: anant.paravastu@chbe.gatech.edu; hzhou43@uic.edu

Among the various amyloid-beta peptides (Aβ), Aβ40 and Aβ42 are the key contributors to Alzheimer's disease (AD)[1]. They can form different types of aggregates including small and large oligomers, protofibrils, and fibrils that can be distinguished based on size and organization[2]. Numerous structural characterizations, mainly by solid-state NMR spectroscopy and cryo-electron microscopy (cryo-EM), have been carried out on Aβ fibrils, both reconstituted in vitro and isolated from patient brains[3–8]. However, a multitude of studies have demonstrated that soluble aggregates like oligomers and protofibrils are more potent inducers of neuronal toxicity and synaptic damage than insoluble fibrils[9–16]. Due to the difficulty in isolating and characterizing the oligomeric species in endogenous aggregates, in vitro Aβ oligomers have been of great interest. A few structural characterizations of Aβ oligomers, by solution and solid-state NMR and cryo-EM, have been reported[17–22]. Very little is known structurally about the assembly pathway of any Aβ aggregate.

Aβ oligomers and fibrils have been produced in cell membranes and mimicking environments[23]. These include ganglioside-rich membranes[24,25], micelles of sodium dodecyl sulfate (SDS)[24,26–28] and dodecylphosphorylcholine (DPC)[20], and synaptic plasma membranes extracted from rat brain tissues[29]. The membranes and mimetics accelerated the fibrillation[24–26] or assisted the nucleation of Aβ40[29], but stabilized small oligomers of Aβ42[20,27]. The oligomers stabilized in SDS micelles were estimated to be in the dimer to tetramer range according to SDS-PAGE electrophoresis[27], whereas those stabilized in DPC micelles were a tetramer according to solution NMR[20]. Upon removal of SDS by dialysis, Aβ42 further grew into a large oligomer[27], which was estimated to have a molecular weight of 150 kD according to light scattering along with size-exclusion chromatography (SEC)[28]. It is not clear how detergent micelles stabilize the small oligomers and how the 150-kD oligomer is formed after detergent removal.

All but one of the over two dozen reported Aβ fibril structures comprise parallel β-sheets[3,5–8]. The exception is the Iowa mutant (D23N) of Aβ40, which can form both parallel and antiparallel fibrils, with the latter only metastable relative to the former[4]. In all cases, each Aβ chain forms two or three β-strands. The DPC-stabilized Aβ42 tetramer of Ciudad et al.[20] contains an antiparallel β-sheet formed by residues Ala30-Ala42; this β-sheet is extended on both edges by a β-strand formed by the N-terminal residues Tyr10-Ala21 of each edge chain, while the N-terminal region (residues 1–29) of the interior chains is disordered. The 150-kD Aβ42 oligomer generated from initial exposure to SDS micelles consists of both a parallel β-sheet and an antiparallel β-sheet, formed by residues Glu11-Val24 and Ala30-Ala42, respectively (Supplementary Fig. 1a)[18,19,21,22]. We refer to these β-strands as N-strand and C-strand, respectively, and the resulting β-sheets as N-sheet and C-sheet. The DPC-stabilized tetramer and the SDS-induced 150-kD oligomer thus share similar C-sheet structures but differ in the N-terminal region.

Here we combine solid-state NMR spectroscopy and molecular dynamics (MD) simulations to investigate the detergent-assisted Aβ42 oligomerization pathway and uncover how SDS and DPC micelles stabilize an Aβ42 tetramer. $^{13}$C-$^{13}$C correlation spectra show that SDS- and DPC-induced large oligomers have the same structure, implying the same oligomerization pathway. Moreover, isotopic dilution in dipolar decay experiments indicates that the tetrameric C-sheet preformed in detergent micelles is directly incorporated into the 150-kD oligomer. Our MD simulations find that this tetrameric C-sheet, comprising antiparallel β-strands, is the only structure stable in SDS micelles. The simulations further reveal that Gly37 and Gly38 destabilize the parallel tetramer but play a key role in stabilizing the antiparallel tetramer. These two

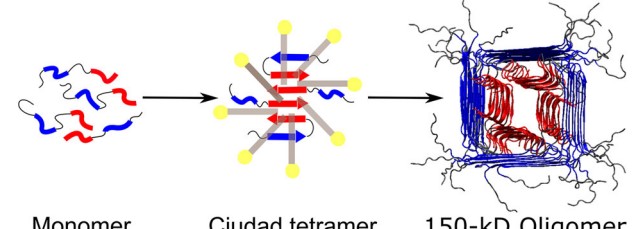

**Fig. 1 Detergent-assisted oligomerization pathway of Aβ42.** Detergent micelles stabilize the antiparallel C-sheet tetramer[20], which is then directly incorporated into the 150-kD oligomer[22]. Blue and red denote N-strands and C-strands, or the residues that would turn into such β-strands.

residues along with Gly33 of the adjacent strand form holes that are filled by the hydrocarbon tails of SDS and DPC molecules.

## Results

We combined solid-state NMR and MD simulations to investigate the detergent-assisted oligomerization pathway of Aβ42. The results lead to the pathway illustrated in Fig. 1, whereby monomers first assemble into a tetramer inside detergent micelles. Burial in the hydrophobic core of detergent micelles stabilizes the tetrameric antiparallel C-sheet. Upon removal of the detergent by dialysis, the C-sheet is directly incorporated into the 150-kD oligomer. Below we present the data underlying this pathway.

As described in more detail below, solid-state NMR experiments were performed on large oligomers formed via initial DPC or SDS stabilization. Both detergents stabilize small oligomers (dimer to tetramer range), which undergo further assembly to large oligomers (~150 kD) upon removal of detergent by dialysis. The similar sizes of the large oligomers from the SDS and DPC preparations were confirmed by SEC. The samples were $^{13}$C-labeled (either uniformly or site-specifically) at selected residues to ensure unequivocal peak assignment.

**DPC-induced large oligomer has the same structure as the SDS-induced 150-kD oligomer.** The solution NMR characterization of Ciudad et al.[20] has shown that Aβ42 forms an antiparallel C-sheet tetramer in DPC micelles. This C-sheet bears much resemblance to the C-sheets that we previously characterized by solid-state NMR on the SDS-induced 150-kD oligomer[18,19,21], including Val36 at the fulcrum of the antiparallel arrangement. In the Ciudad et al. structure, the Val36 residues in the two central C-strands align with Met35 of an edge strand on either side.

Following our previous solid-state NMR characterization of the SDS-induced 150-kD oligomer[18,19,21,22], we carried out $^{13}$C-$^{13}$C dipolar assisted rotational resonance (DARR) measurements[30,31] on SDS- and DPC-induced large oligomers to probe distances up to 6 Å between labeled sites. The $^{13}$C-$^{13}$C correlation spectra acquired with a 50 ms mixing time on samples with uniform $^{13}$C-labeling at Glu11, Lys16, Phe19, and Val36 are shown in Fig. 2a, with black contours for the SDS-induced oligomer and red contours for the DPC-induced oligomer. With a 50 ms mixing time, DARR measurements capture crosspeaks of $^{13}$C sites within the same residue; these intra-residue crosspeaks are sensitive to secondary, tertiary, and quaternary structures. The near perfect match between the two spectra, with a root-mean-squared deviation (RMSD)[32] of only 0.15 ppm over the range of 10 to 75 ppm in both dimensions, demonstrates that the DPC-induced oligomer has the same structure as the SDS-induced oligomer. Supplementary Fig. 2 presents alternative displays of

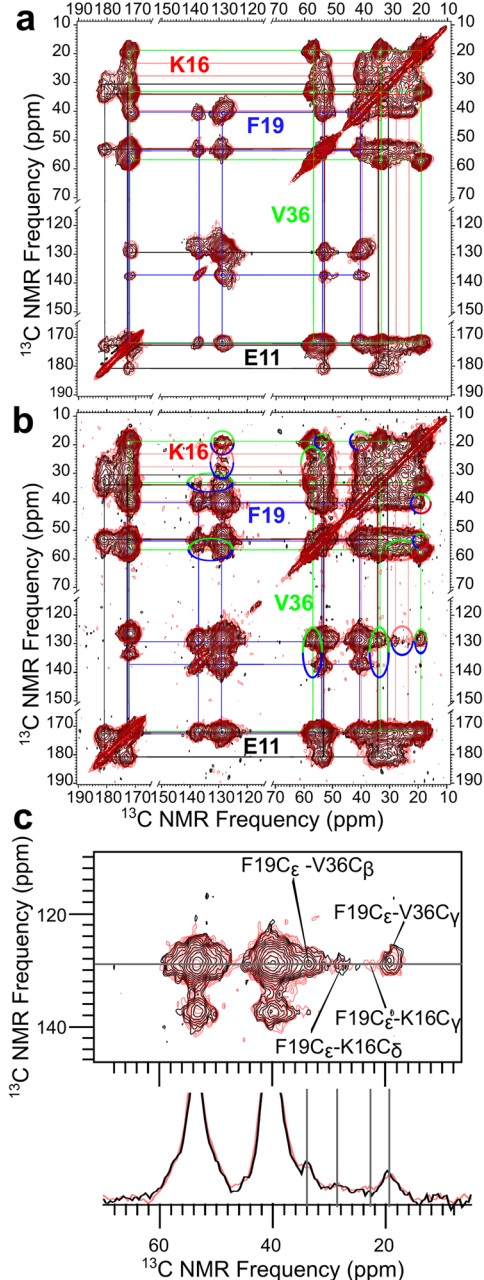

**Fig. 2 SDS- and DPC-induced large oligomers have the same structure.**
**a** Overlaid $^{13}C$-$^{13}C$ correlation spectra of SDS- and DPC-induced oligomers (black and red contours, respectively), acquired at a 50 ms mixing time on samples with uniform $^{13}C$-labeling at Glu11, Lys16, Phe19, and Val36. The colored lines indicate peaks connected by intra-residue $^{13}C$-$^{13}C$ couplings. **b** Overlay of the correlation spectra acquired on the same samples but with a 500 ms mixing time. The multi-colored ovals indicate crosspeaks corresponding to $^{13}C$ atoms on different residues. The spectrum of the SDS-induced 150-kD oligomer (black contours) was published previously[21]. **c** A zoom highlighting the crosspeaks of Phe19 aromatic carbons with Lys16 and Val36 aliphatic carbons, along with a 1D slice illustrating the close match of the two spectra.

the $^{13}C$-$^{13}C$ correlation spectra as well as 1D slices to highlight their near perfect match.

We also performed DARR measurements on these two oligomer samples with a longer, 500 ms mixing time (Fig. 2b), allowing for the detection of crosspeaks between different residues. The 500-ms spectra exhibit crosspeaks between all

combinations of Lys16, Phe19, and Val36. Figure 2c presents a zoomed view into the crosspeaks of Phe19 aromatic carbons with Lys16 and Val36 aliphatic carbons, along with a 1D slice illustrating the close match of the two spectra. As reported previously[21], the Lys16-Phe19 crosspeaks arise because these residues are located in parallel N-strands (residues Glu11 to Val24) with alternating registry shifts; the Lys16-Val36 and Phe19-Val36 crosspeaks arise from sidechain-sidechain contacts between the N- and C-sheets. Additional data including DARR restraints and cryo-EM images have led to a structural model for the 150-kD oligomer[22], shown schematically in Fig. 1.

The observation that the DPC- and SDS-induced large oligomers have the same structure and the fact that these two types of detergents have the same hydrocarbon tail (Supplementary Fig. 3) strongly suggest that the same oligomerization pathway is followed, including the same small oligomeric species inside the detergent micelles. That is, the tetramer found by Ciudad et al.[20] in DPC micelles also represents the small oligomeric species in SDS micelles. Ciudad et al. did not observe other oligomers, such as ones where the N-sheet instead of the C-sheet is buried in micelles. As presented below, our MD simulations find that the antiparallel C-sheet tetramer is the only small oligomer that is stable in SDS micelles.

**SDS-stabilized small oligomers form an antiparallel C-sheet that is directly incorporated into the 150-kD oligomer.** The antiparallel C-sheets, with Val36 at the fulcrum, form the core of the 150-kD oligomer (Fig. 1)[18,19,21,22]. This special role of Val36 was established by the PITHIRDS-CT experiment[33], which measures the decay of $^{13}C$ signals due to dipolar recoupling, on a sample with $^{13}C$ on the carbonyl carbon of Val36. The resulting PITHIRDS-CT decay curve can be interpreted as indicating $^{13}C$-$^{13}C$ distances within 5 Å, i.e., with Val36 $^{13}CO$ sites placed on neighboring strands in the C-sheets of the 150-kD oligomer.

To uncover the time point when the C-sheets are formed, we applied isotopic dilution at different stages in the oligomer preparation protocol (Fig. 3a). At two different stages, Val36 $^{13}CO$-labeled and unlabeled components were mixed at a 1:1 ratio to produce two different 150-kD oligomer samples with isotopic dilution. We also extended isotopic labeling to Ala30 $^{13}C_{\beta}$ as a negative control (see below). In the "pre-SDS" sample, labeled and unlabeled monomers were mixed before being introduced into SDS micelles. In the "post-SDS" sample, two species of SDS-stabilized small oligomers were mixed before dialysis: one had the isotopic labeling and the other did not. Relatively to the 100% Val36 $^{13}CO$-labeled oligomer, both isotopically diluted samples have reduced probabilities of placing Val36 $^{13}CO$ sites on neighboring β-strands in the C-sheets. However, the post-SDS sample has a higher probability of doing so, if the small oligomers are directly incorporated into the 150-kD oligomer, i.e., without first disassembling into monomers.

As shown in Fig. 3b, the PITHIRDS-CT decays of Val36 $^{13}CO$ in both 150-kD oligomer samples with isotopic dilution are less rapid than the counterpart with 100% Val36 $^{13}CO$ labeling, due to reduced probabilities of Val36 $^{13}CO$ sites being on neighboring strands of the C-sheets. However, between the two isotopically diluted samples, the PITHIRDS-CT decay is more rapid when isotopic dilution occurred after small oligomer formation in SDS micelles. A fit to a near-Lorentzian function yielded decay times of 36 ± 1 ms for the pre-SDS sample and 30 ± 1 ms for the post-SDS sample. The more rapid decay in the latter sample can happen if the C-sheet preformed in SDS micelles is directly incorporated into the 150-kD oligomer.

To rule out the possibility that isotopic dilution just somehow impedes PITHIRDS-CT decays in general, we added Ala30 $^{13}C_{\beta}$

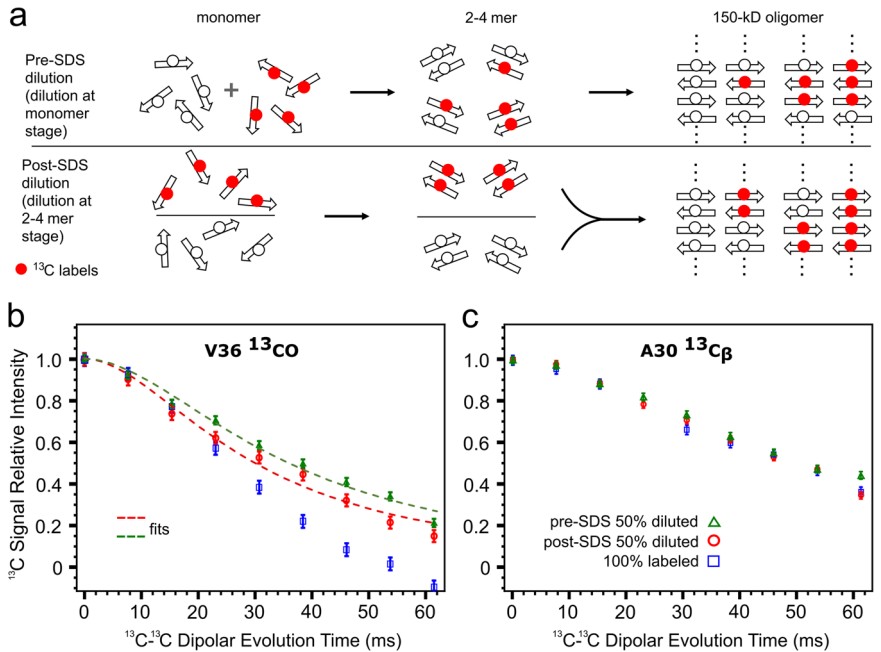

**Fig. 3 Effects of isotopic dilution demonstrate direct incorporation of antiparallel C-sheet tetramers into the 150-kD oligomer. a** Illustration of isotopic dilution at two different stages of the oligomer preparation protocol. In the hypothetical scenario where monomers combine to form dimers that then combine to form tetramers, with isotopic dilution at the monomer stage, only 50% of the tetramers would have two or more neighboring V36 $^{13}$CO sites. In comparison, with isotopic dilution at the dimer stage, 75% of the tetramers would have two or more neighboring V36 $^{13}$CO sites, leading to a more rapid PITHIRDS-CT decay. **b** PITHIRDS-CT decays of V36 $^{13}$CO in the 100% labeled oligomer and in the pre-SDS diluted and post-SDS diluted samples. **c** Corresponding results for Ala30 $^{13}$C$_\beta$ in the same samples. The data points in **b, c** represent the peak heights from a nonlinear fit of NMR peaks to a Gaussian function using Mathematica's NonlinearModelFit function. The error bars represent 95% confidence intervals calculated by Mathematica. In **b**, the decays of V36 $^{13}$CO in the pre-SDS diluted and post-SDS diluted samples were fit to a near-Lorentzian function, $y = 1/[1 + (t/t_d)^\alpha]$ with $\alpha = 1.84$ (green and red curves).

labeling along with Val36 $^{13}$CO labeling in these samples as a negative control. In antiparallel C-sheets, Ala30 $^{13}$C$_\beta$ sites are at least 10 Å apart and hence isotopic dilution should have minimal effects on PITHIRDS-CT decays. Indeed, as shown in Fig. 3c, the PITHIRDS-CT decay curves of Ala30 $^{13}$C$_\beta$ in the two diluted samples are hardly indistinguishable from that of the 100% labeled sample. The PITHIRDS-CT decay curves of both V36 $^{13}$CO and A30 $^{13}$C$_\beta$ in the 100% labeled sample are similar to those that we published 10 years ago using different samples[18]. Here, we took great care to minimize sample-to-sample variations by following exactly the same preparation procedure for the three samples in Fig. 3b, c. The close superposition of the decays curves of A30 $^{13}$C$_\beta$ in the three samples shown in Fig. 3c thus demonstrates minimal sample-to-sample variations. Because the decay curves of V36 $^{13}$CO shown in Fig. 3b were measured on the same three samples as in Fig. 3c, we can be confident that the difference in decay times, $36 \pm 1$ ms for the pre-SDS sample and $30 \pm 1$ ms for the post-SDS sample, is beyond sample-to-sample variations. Thus we can conclude that the slowed PITHIRDS-CT decays of Val36 $^{13}$CO in both of the isotopically diluted samples, relative to the 100% labeled sample, are due to its special position at the fulcrum of the antiparallel C-sheets. Moreover, the simplest interpretation for the faster decay in the post-SDS sample, relative to the pre-SDS sample, is direct incorporation of SDS-stabilized C-sheets into the 150-kD oligomer (Fig. 3a).

**N-sheet dimer and tetramer and C-sheet dimers are unstable in SDS micelles.** To confirm that the small oligomers formed in SDS micelles are a tetramer with an antiparallel C-sheet and provide atomistic insight into how detergent micelles assist the oligomerization of Aβ42, we carried out all-atom MD simulations of

different forms of Aβ dimers and tetramers (Supplementary Fig. 1) in SDS micelles. SDS micelles have an estimated aggregation number in the range of 62 to 101[34]. In agreement, in our MD simulations, pure SDS micelles comprising 60, 80, and 100 molecules remained intact as a single micelle, whereas micelles with 120 and 150 molecules split into two smaller micelles (Supplementary Fig. 4). Preliminary simulations with 99 to 114 SDS molecules surrounding Aβ dimers or tetramers also showed splitting of the micelles. In subsequent preparation of the systems with Aβ dimers and tetramers buried in SDS micelles, we limited the number of SDS molecules to the range of 59 to 80. MD simulations of each system were run in four replicates, each for at least 1000 ns (Supplementary Table 1).

The C-strand (residues 30-42) consists entirely of nonpolar residues, but half of the N-strand (residues 11-24) comprises charged and polar residues. The latter composition makes the hydrophobic core of detergent micelles an unfavorable solubilization environment for N-sheets. Indeed, in MD simulations where a parallel N-sheet dimer or tetramer was initially placed in the core of an SDS micelle (Fig. 4a, b), the micelle quickly (within 10 ns) breaks open, exposing the N-sheet (Fig. 4c). The ends of the N-strands then start fraying and separating from each other (Fig. 4d, e). Correspondingly, the number of backbone hydrogen bonds steadily decreases (Supplementary Fig. 5a, b).

Next we considered the possibility that the small oligomeric species inside SDS micelles is a C-sheet dimer. We prepared two Aβ42 dimer models, one with a parallel C-sheet buried at the center of an SDS micelle, and the other is similar but with an antiparallel C-sheet (Fig. 5a, b). For both models, the ends of the C-strands start fraying within 20 ns, and the β-sheets move to the surface of the micelle by 100 ns. The C-strands continue to fray and separate from each other (Fig. 5c, d), resulting in a steady

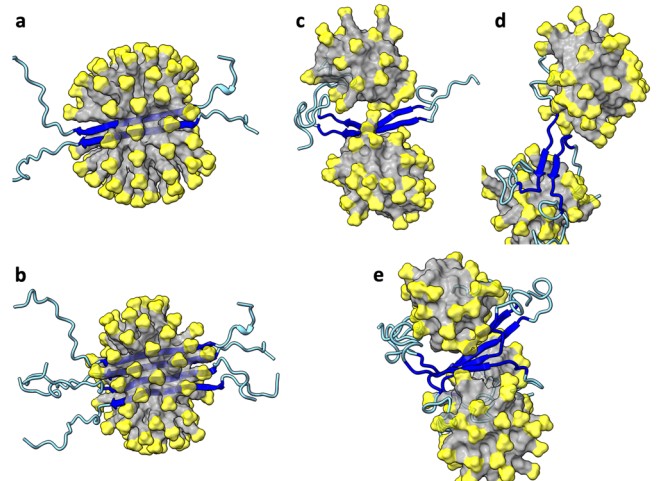

**Fig. 4 MD simulations of the Aβ42 parallel N-sheet dimer and tetramer in SDS micelles. a** Initial structure of dimer. **b** Initial structure of tetramer. **c** Dimer at 10 ns of a simulation. **d** Dimer at 601.6 ns. **e** Tetramer at 939.3 ns. The SDS micelles are shown in surface representation with headgroups in yellow and hydrocarbon tails in grey. Aβ42 molecules are shown in cartoon representation with residues 1-10 and 25-42 in cyan and residues 11-24 in blue.

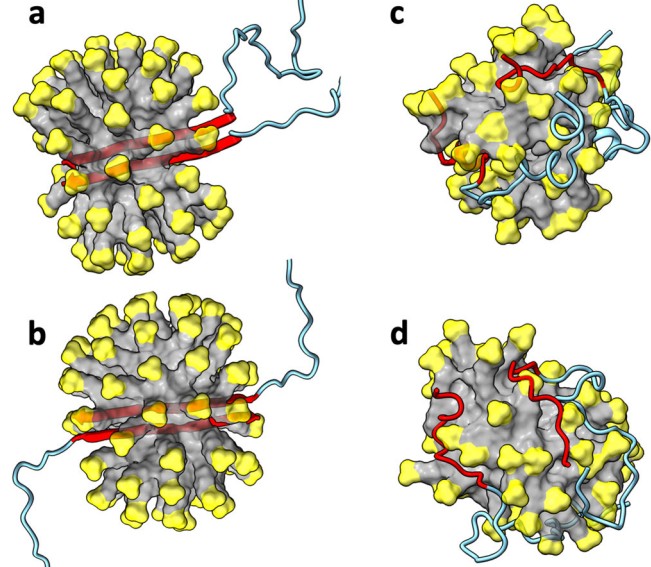

**Fig. 5 MD simulations of the Aβ42 C-sheet dimers in SDS micelles.**
**a** Initial structure of parallel dimer. **b** Initial structure of antiparallel dimer. **c** Parallel dimer at 947.1 ns of a simulation. **d** Antiparallel dimer at 661.3 ns. The SDS micelles are shown in surface representation with headgroups in yellow and hydrocarbon tails in grey. Aβ42 molecules are shown in cartoon representation with residues 1 to 29 in cyan and residues 30-42 in red.

decline in backbone hydrogen bonds (Supplementary Fig. 5c, d). Interestingly, the N-terminal region frequently forms α-helices on the micellar surface (e.g., Fig. 5c). Very similar results are obtained for an Aβ40 dimer with a parallel C-sheet (Supplementary Figs. 6 and 5e). These observations indicate that a dimeric C-sheet has too few backbone hydrogen bonds to maintain its stability in an SDS micelle, and point to a larger oligomer, i.e., tetramer, as the species stabilized by SDS micelles in our oligomer preparation.

**Parallel C-sheet tetramers are only partially stable in SDS micelles.** While the Aβ42 C-sheet is antiparallel both in the tetramer structure of Ciudad et al.[20] and in our 150-kD oligomer[18,19,21,22], parallel C-sheets are observed in nearly all Aβ fibril structures. We tested whether Aβ parallel C-sheet tetramers are stable in SDS micelles (Fig. 6a, b). As soon as 10 ns into the simulations, the Aβ42 parallel tetrameric C-sheet develops a bend around Gly37 and Gly38. By 50 ns, the C-terminal region starting at Gly37 comes out of the micelle to become exposed to water. Thereafter the C-terminal region continues to experience perturbations, with some fraying and one strand even separating in one of the four replicate simulations (Fig. 6c). Correspondingly, the number of backbone hydrogen bonds exhibits null to modest decline over the simulation time (Supplementary Fig. 5f).

The situation is similar for the Aβ40 parallel C-sheet tetramer, except that the C-terminal region starting from Gly37 is almost completely frayed and separated between chains (Fig. 6d). The shorter parallel C-sheet of Aβ40 thus suffers a significant decline in the number of backbone hydrogen bonds (Supplementary Fig. 5g); the last two residues of Aβ42, Ile41 and Ala42, evidently provide some resistance to fraying. The bending and fraying of these Aβ tetramers can be attributed to the two consecutive flexible residues, Gly37 and Gly38, which are aligned across a parallel β-sheet (Supplementary Fig. 7a).

**Aβ42 antiparallel C-sheet tetramer is stable in both SDS and DPC micelles.** In contrast to the instability of the foregoing Aβ models, the Aβ42 antiparallel C-sheet tetramer is very stable in

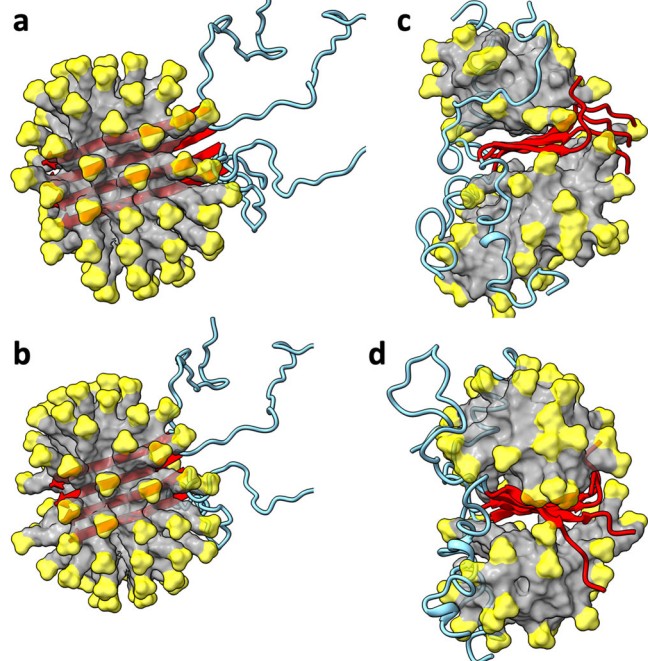

**Fig. 6 MD simulations of the Aβ parallel C-sheet tetramers in SDS micelles. a** Initial structure of Aβ42 tetramer. **b** Initial structure of Aβ40 tetramer. **c** Aβ42 tetramer at 834.2 ns of a simulation. **d** Aβ40 tetramer at 994.2 ns. The SDS micelles are shown in surface representation with headgroups in yellow and hydrocarbon tails in grey. Aβ molecules are shown in cartoon representation with residues 1 to 29 in cyan and residues 30-42/40 in red.

SDS micelles in our simulations. In a micelle comprising 70 SDS molecules, the antiparallel C-sheet maintains its initial planar structure (Fig. 7a, b). The backbone hydrogen bonds are essentially intact for the entire simulations of 2200 ns (Supplementary

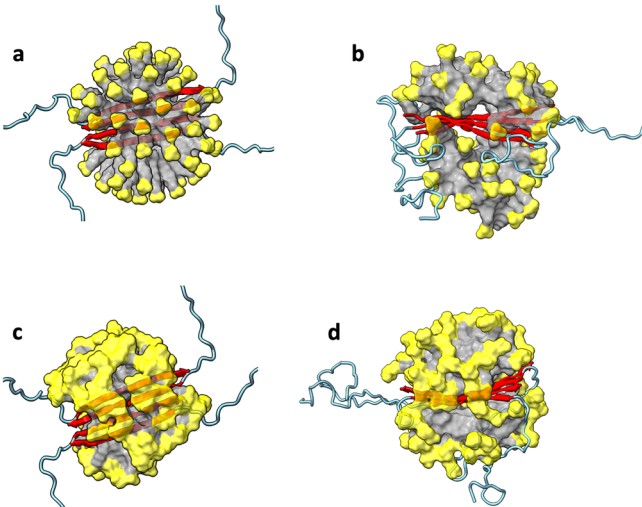

**Fig. 7 MD simulations of the Aβ42 antiparallel C-sheet tetramers in SDS and DPC micelles. a** Initial structure in an SDS micelle. **b** Snapshot in SDS at 869.3 ns of a simulation. **c** Initial structure in a DPC micelle. **d** Snapshot in DPC at 500.0 ns. The micelles are shown in surface representation with headgroups in yellow and hydrocarbon tails in grey. Aβ42 molecules are shown in cartoon representation with residues 1 to 29 in cyan and residues 30-42 in red.

Fig. 5h). The Aβ42 antiparallel C-sheet tetramer is also stable in another set of simulations in a micelle comprising 61 SDS molecules. Moreover, this tetramer is also stable in DPC micelles, whether comprising 71 (Fig. 7c, d and Supplementary Fig. 5i) or 61 DPC molecules.

That the parallel C-sheet becomes bent around Gly37 and Gly38 but the antiparallel C-sheet remains planar can be explained by the fact that these consecutive, flexible residues are distributed in opposite ends of the C-sheet due to the antiparallel arrangements of the C-strands (Supplementary Fig. 7b). In essence, the intervening C-strands, with non-Gly residues aligned with Gly37 and Gly38 of adjacent C-strands by backbone hydrogen bonds, prevent chain bending.

Taken together, our MD simulations show that the antiparallel C-sheet tetramer is likely the only Aβ42 small oligomer that is stable in SDS and DPC micelles. This is also the only structure observed by Ciudad et al.[20]. That the antiparallel tetrameric C-sheet is the only stable species in detergent micelles provides a strong basis for our interpretation of the isotopic dilution data as indicating direct incorporation of this preformed structure into the 150-kD oligomer.

**Detergent hydrocarbon tails fill glycine holes to stabilize antiparallel C-sheet**. Although the antiparallel arrangement prevents the bending of the C-strands, Gly37 and Gly38 would still create voids on both sides of the C-sheet, due to the lack of sidechains (Fig. 8a). In fact, Gly38 and Gly33 are aligned within each half of the tetramer, therefore further extending the voids; we refer to them as glycine holes. There are four distinct types of glycine holes, labeled as I, II, III, and IV; each type appears twice in the tetramer. Types I and II are on one face of the C-sheet, with either Gly37/Gly38 or Gly33 on an edge strand. Types III and IV are on the opposite face of the C-sheet, with either Gly33 or Gly37/Gly38 on an edge strand.

Remarkably, detergent molecules fill the glycine holes with their hydrocarbon tails, which then interact with surrounding aliphatic sidechains (Fig. 8b). In Supplementary Fig. 8a, we display the probabilities that the eight glycine holes are filled with

a detergent molecule in simulations of the antiparallel C-sheet tetramer in SDS and DPC micelles, each at two sizes. The type II and type III holes are filled with close to 30% probabilities whereas the type I and type IV holes are filled with ~10% probabilities. Type II is favored because the detergent hydrocarbon tail can interact with Ile31 and Met35 on the first strand (i.e., edge strand) and Val36 and Val40 on the second strand, but also with Leu34 on the third strand (Fig. 8b, top panel). In contrast, for type I, Leu34 on the second strand prevents the hole from extending to the third strand. The type III holes can be filled in a similar way as the type I holes, with the detergent tail inserted from the edge strand and extending to Met35 of the third strand. However, whereas a type II hole is separated, by Leu34 on the third strand, from a type I hole on the opposite edge, on the flip side of the C-sheet, a type III hole is connected with a type IV hole on the opposite edge (Fig. 8a, bottom panel). As a result, a type III hole is most frequently filled by a detergent tail coming through the connected type IV hole, interacting with Ile32 and Leu34 on the first strand, Val39 on the second strand, and Met35 on the third strand (Fig. 8b, bottom panel).

In Supplementary Fig. 8b, we display the probabilities that a certain number of the glycine holes are filled at any given time. The probabilities closely follow a binomial distribution, indicating that detergent molecules fill the glycine holes independently. In ~85% of the simulation time, at least one of the glycine holes is filled with a detergent molecule; in more than half the time, two or more holes are filled. These hole-filling detergent molecules provide significant stabilization to the Aβ42 antiparallel C-sheet tetramer.

**N-terminal region forms an additional β-strand docked to the antiparallel C-sheet in detergent micelles**. In our initial structure of the Aβ42 antiparallel C-sheet tetramer, we modeled the N-terminal region (residues 1-29) as unstructured. In the simulations in the micelle with 61 DPC molecules, the N-terminal region of an edge chain readily forms an additional β-strand that is docked to the edge strand of the C-sheet (Supplementary Fig. 9a and Supplementary Movie 1). In some simulations, the docking of the N-terminal region near the C-sheet is initiated by Lys16 contacting a DPC headgroup, and stabilized by the burial of Leu17 (Supplementary Fig. 9b and Supplementary Movie 2). In other simulations, the docking is initiated by Lys28. The addition of the N-terminal β-strand was also observed in the micelles with 71 DPC molecules or with 61 and 70 SDS molecules (Supplementary Fig. 10). This N-terminal β-strand was found in the tetramer structure of Ciudad et al.[20] determined in DPC micelles. The formation of this additional β-strand in our simulations provides strong validation of the simulations.

In addition to the formation of the additional β-strand, the simulations also show that the N-terminal region can form an α-helix on the micellar surface (Supplementary Movie 1). Such α-helix formation was reported by Ciudad et al.[20] based on secondary chemical shifts, providing additional validation. N-terminal α-helix formation on the micellar surface is observed in most of our simulations.

**Discussion**

We have combined solid-state NMR spectroscopy and molecular dynamics simulations to investigate the detergent-assisted oligomerization pathway of Aβ42. SDS and DPC micelles stabilize the antiparallel C-sheet tetramer, which is then directly incorporated into the 150-kD oligomer. These observations provide crucial information on the oligomerization pathway (Fig. 1). The detergents induce the formation of the tetrameric C-sheet, which in water would be an extremely slow process. We can speculate that,

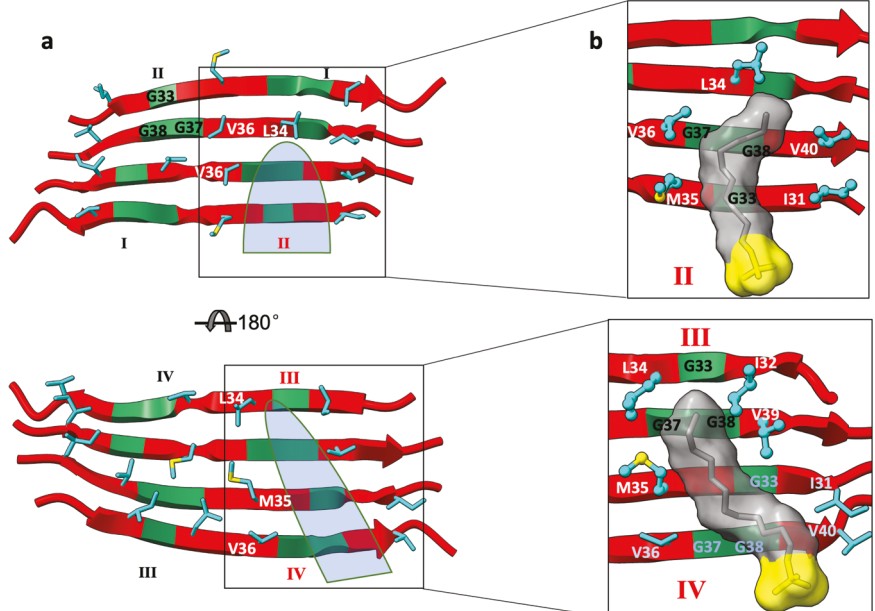

**Fig. 8 Glycine holes and their filling by detergent molecules. a** Types of glycine holes. Top panel and bottom panels show views into the antiparallel C-sheet from opposite directions. Whereas a type II hole (cyan shading) is separated by Leu34 from a type I hole, a type III hole is connected with a type IV hole (cyan shading). **b** Enlarged views into how an SDS molecule fills a type II hole (top panel) or a type III hole (bottom panel). C-strands are shown in cartoon representation with G37, G38, and G33 in green and non-glycine residues in red. Sidechains lining the glycine holes are shown in stick (carbon in cyan and sulfur in yellow); the representation switches to ball-and-stick if they interact with an SDS molecule. SDS molecules are shown in both stick and surface representations, with the headgroup in yellow and the hydrocarbon tail in grey.

once formed, multiple copies of the tetrameric C-sheet stack laterally to allow for the formation of the N-sheet. Both monomers and preformed C-sheet tetramers then stack longitudinally to grow the oligomer into the 150-kD size. Indeed, our MD simulations have already shown that the preformed C-sheet can template the formation of the N-strand in the micellar environment. Based on our structural model (illustrated schematically in Fig. 1), we have speculated a reason why the oligomer stops growing at 150 kD[22]. The N- and C-strands of protomers in the oligomer adopt two shapes, "L" and "U". While maximum oligomeric stability is achieved when the two types of protomers are staggered in perfect order, the actual assembly process may deviate from this perfect order, thereby generating defects. Such defects are favored from entropic consideration but reduce oligomer stability. The balance between entropy and stability determines the oligomer size.

The completely nonpolar composition of the C-strand provides the necessary condition for SDS and DPC micelles to stabilize C-sheet tetramers. However, our study has further revealed important roles of the three C-strand Gly residues. Gly37 and Gly38 weaken the stability of the parallel C-sheet tetramer, thereby eliminating an alternative pathway of detergent-assisted oligomerization. This observation also explains why our SDS-based preparation protocol does not produce Aβ40 oligomers[26], since Aβ40 is only known to form a parallel C-sheet and such a sheet is unstable in detergent micelles especially due to the missing two C-terminal residues. For the Aβ42 antiparallel C-sheet tetramer, the three C-strand Gly residues form holes that are filled by the hydrocarbon tails of the detergent molecules, thereby turning a potentially destabilizing feature into a stabilizing factor.

Our study suggests ideas for designing sequences that follow a specific aggregation pathway. In particular, if Gly37 and Gly38 are mutated into an amino acid, e.g., Val, that is favored in β-strands, then detergent micelles may stabilize the parallel C-sheet tetramer of Aβ42 (or even Aβ40). In MD simulations, parallel

C-sheet tetramers carrying the G37V and G38V mutations are indeed stable in SDS micelles (Supplementary Fig. 11). They may then seed the formation of fibrils instead of oligomers. Conversely, with the Iowa mutation (D23N), an antiparallel N-sheet of Aβ40 could become stable enough in detergent micelles, which could then seed the formation of oligomers instead of fibrils.

Structural studies have focused on defining the final oligomers or fibrils but very little is known about their assembly pathways. The present study has defined crucial structural characteristics of one Aβ42 oligomerization pathway. The interplay of the Aβ42 primary, secondary, and tertiary structures with the solubilization environment revealed here is instructive for understanding other assembly pathways. In particular, cellular surfaces encountered by endogenous Aβ peptides provide numerous opportunities for assisting aggregation.

## Materials and methods

**Peptide synthesis and oligomer sample preparation.** Isotope-labeled compounds used in the syntheses were purchased from Cambridge Isotope Laboratories. Labeled and unlabeled Aβ42 peptides were synthesized by the Proteomics Core at the Mayo Clinic.

Chemicals used for preparing oligomer samples were purchased from Sigma Aldrich. The crude product from the synthesis of Aβ42 peptides was dried using hexafluoro isopropanol. It was then dissolved in 0.1 M NaOH and subjected to SEC using a Superdex 75HR 10/30 column (Amersham Pharmacia). The column was equilibrated with 20 mM sodium phosphate buffer (pH 7.4) at a flow rate of 0.5 ml/min. This step aimed to isolate Aβ42 monomer.

Small oligomers (in the dimer to tetramer range[27]) were prepared by incubating aliquots of the SEC-purified Aβ42 monomer overnight at room temperature in the presence of either 4 mM SDS and 50 mM sodium chloride or 2 mM DPC. These small oligomers were then subjected to dialysis against

20 mM sodium phosphate buffer for 48-72 hours, with three to eight buffer changes. Subsequently, it was dialyzed against 10 mM sodium phosphate buffer for 3-4 hours to remove detergents and decrease the salt concentration. The resulting samples of large oligomers (~150 kD) were filtered using an Amicon Ultra 4 centrifugal concentration/filtration device with a molecular mass cutoff of 50 kD to remove monomers and small oligomers. Large-oligomer samples were lyophilized before being loaded into NMR rotors (sample holders) for solid-state NMR measurements.

**Solid-state NMR spectroscopy.** All solid-state NMR experiments on SDS- and DPC-induced oligomer samples were performed on a 500 MHz Bruker spectrometer, equipped with a 3.2 mm HCN MAS (magic angle spinning) probe and a 3.2 mm double-resonance MAS probe. $^{13}C$-$^{13}C$ DARR measurements were made at an MAS rate of 11 kHz and a short mixing time of 50 ms and long mixing time of 500 ms to identify intra-residue and inter-residue crosspeaks, respectively. Crosspeak positions were determined by fitting to a 2D Gaussian function. Partial peak assignments were made using a custom Mathematica code and known $^{13}C$ chemical shifts from the Biological Magnetic Resonance Bank (https://bmrb.io/).

The RMSD value between two $^{13}C$-$^{13}C$ correlation spectra was determined according to Qiang et al.[32]. The procedure includes the following ingredients: (1) the two spectra were interpolated into a common 2D grid (with 0.2 ppm spacing in each dimension); (2) only grid points where the signal intensities were above four times the noise level (estimated by the intensities in regions free of crosspeaks) in at least one of the two spectra were counted; (3) the intensities of one spectrum were scaled by a constant factor to minimize mismatch in intensities.

PITHIRDS-CT measurements were made at an MAS rate of 12.5 kHz and a dipolar recoupling time between 0 to 61.4 ms[33]. During PITHIRDS-CT recoupling and acquisition, proton decoupling at 100 kHz was carried out using continuous wave decoupling. Signal averaging of 24 h was used for each PITHIRDS-CT experiment.

**System preparation for MD simulations.** The SDS micelles were prepared using the micelle maker web server[35]. For the SDS-only simulations, five systems with different numbers of SDS (60, 80, 100, 120, and 150 SDS molecules) were built.

Eight Aβ dimers or tetramers were prepared (Supplementary Fig. 1 and Supplementary Table 1). The initial structure for the N-sheet (formed by residues 11-24) was from the solid-state NMR structure 6TI6[6]; the initial structure for the parallel and antiparallel C-sheets (residues 30-42/40) were from the solution NMR structures 2BEG[3] and 6RHY[20], respectively. The rest of the peptide chains were modeled as disordered and generated using TRADES[36]; the β-sheet and disordered regions were combined to form full-length Aβ oligomers using VMD[37]. The Aβ dimers or tetramers were placed in the center of a micelle with 80-120 SDS molecules, and overlapping SDS molecules within 1.6 Å were removed using VMD. For the Aβ42 antiparallel C-sheet tetramer, this last step resulted in a micelle with 70 SDS molecules. Removal of additional SDS molecules within 2.6 Å of the Aβ42 tetramer resulted in another system with 61 SDS molecules.

The Aβ42 antiparallel C-sheet tetramer in DPC micelles was prepared in a similar way, but with SDS molecules in the initial micelles all replaced with DPC molecules. The initial structure of a single DPC molecule was generated by the CHARMM-GUI web server[38]. An initial micelle with 120 SDS molecules, after replacing with DPC and removing overlapping DPC molecules within 2.6 Å, produced a system with the Aβ42 surrounded by 71

DPC molecules. A second preparation, with the initial SDS count reduced to 100 and the distance for removing overlapping DPC molecules reduced to 1.6 Å, produced a system with the Aβ42 surrounded by 61 DPC molecules.

The pure SDS micelles were solvated in a 100 Å × 100 Å × 100 Å water box generated from the CHARMM-GUI web server; overlapping water molecules within 1.5 Å were removed using VMD. The dimensions for the Aβ-containing systems were adjusted to 100 Å × 100 Å × 120 Å for all the six parallel dimers and tetramers, to 80 Å × 120 Å × 140 Å for the antiparallel dimer and for the antiparallel tetramer surrounded by 70 SDS molecules, and to 100 Å × 130 Å × 160 Å for antiparallel tetramer surrounded by 61 SDS molecules or 61 or 71 DPC molecules. Na$^+$ and Cl$^-$ ions were added to each system using VMD[37] for charge neutralization and generating a salt concentration of 30 mM.

**MD simulations.** MD simulations were run in AMBER18[39] using the CHARMM36m[40] force field for proteins and micelles and TIP3P model[41] for water. This force field combination has been shown to work well for SDS-containing systems[42]. After energy minimization (2500 cycles of steepest descent followed by 2500 cycles of conjugate gradient), each system was equilibrated in six steps. The first two were at constant temperature and volume while the remaining four were at constant temperature and pressure. The timesteps were 1 fs in the first three and increased to 2 fs in the last three. The simulation times in the six steps were 125 ps, 125 ps, 125 ps, 2000 ps, 500 ps, and 500 ps for the SDS-only systems and were 125 ps, 500 ps, 500 ps, 1000 ps, 1000 ps, and 1000 ps for the Aβ-SDS/DPC systems. During these six steps, restraints on the SDS sulfur atom or the DPC phosphorus atom were gradually reduced with the force constant changing from 2.5 kcal/mol/Å$^2$ to 0; similarly, the force constant on protein heavy atoms was reduced from 10 kcal/mol/Å$^2$ to 0. Production runs were carried out at constant temperature and pressure without restraints on GPUs using *pmemd.cuda*[43], with total simulation times listed in Supplementary Table 1.

Bond lengths containing hydrogen were constrained using the SHAKE algorithm[44]. Long-range electrostatic interactions were treated using the particle mesh Ewald method[45] with a nonbonded cutoff of 12 Å. The temperature was maintained at 300 K by the Langevin thermostat[46] with a damping constant of 1 ps$^{-1}$; the pressure was maintained at 1 bar by the Monte Carlo barostat[47]. Snapshots were saved every 100 ps for analysis.

**MD data analysis.** The number of hydrogen bonds was calculated using the *hbond* plugin in CPPTRAJ[48], with the distance and angle cutoffs at 3.5 Å and 125°, respectively. Line plots were made in Python 3.9.7, with smoothing by a Savitzky-Golay filter[49] from the *SciPy* library, over a window of 501 points.

Filling of glycine holes was detected by running a tcl script in VMD. A hole was considered filled if any of the carbon atoms in the tail of a detergent molecule was within 5 Å of both (1) at least one backbone heavy atom on each of Gly33, Gly37, and Gly38 and (2) Cβ of at least one of the sidechains lining the particular glycine hole (Fig. 8a). After removing the first 10 ns, the fraction of saved snapshots in which the hole was filled was taken as the probability for that event; this probability was averaged over the four replicate simulations.

A binominal distribution was calculated by assuming each of the eight glycine holes was filled independently and with the same probability. The latter probability was taken as the average of the probabilities for filling the eight holes obtained from the simulations.

**Statistics and reproducibility**. MD simulations for each system were run in four replicates, each for at least 1000 ns. The results from the replicates were very similar, demonstrating reproducibility. Convergence of the MD simulations was verified by calculating the mean numbers of hydrogen bonds in blocks of 250 ns along a single trajectory; the variance along a single trajectory was no more than the variance among replicate trajectories.

**Reporting summary**. Further information on research design is available in the Nature Portfolio Reporting Summary linked to this article.

## Data availability

All data generated or analyzed during this study are included in this published article (and its supplementary data files). The source data for all the plots presented in figures are deposited in GitHub at https://github.com/hzhou43/Abeta_in_detergents.

## Code availability

Data analysis procedures were described under Materials and Methods. All the computer programs used were cited and are publicly available. The input files for MD simulations and the initial and final coordinate files are deposited in GitHub at https://github.com/hzhou43/Abeta_in_detergents.

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

## Acknowledgements
This work was supported by National Institutes of Health Grant AG073434.

## Author contributions
H.-X.Z., T.L.R., and A.K.P. designed the research; F.N.K.M., R.P., Y.G., T.R.S., A.S.R., and J.O.W. performed the research and analyzed the data; F.N.K.M. and H.-X.Z. wrote the manuscript with input from A.K.P.

## Competing interests
The authors declare no competing interests.
