## [Peer Review File · Communications Biology]

Reviewers' comments:

Reviewer #1 (Remarks to the Author):

In the study by Zhou and colleagues, the prevalent pathways of A β 42 oligomerization in the presence of the detergents, SDS and DPC, were explored. The authors observed that A β 42 monomers cluster into tetramers that consist of antiparallel C-sheets. These tetramers gain stability from the interactions between GLY 33, 37, 38 and the detergent's hydrocarbon tails. With the support of SDS/DPC micelles, these A β 42 tetramers are fortified, and the elimination of SDS subsequently results in the emergence of 150-kD A β 42 oligomers. These insights are valuable for understanding A β oligomer aggregation in cellular environments and the formation of oligomers or fibrils influenced by specific sequences or residues. The paper is clearly presented, and the findings are systematically outlined. To enhance the quality of the work, I recommend addressing the following points:

1. It's worth noting that the tetramers, stabilized by either SDS or DPC and consisting of antiparallel C-sheets, could transition directly into larger oligomers. The tetramer structure highlighted in the study by Ciudad et al. showcases four antiparallel C-sheets alongside two N-sheets. This raises an intriguing query about the role of the N-sheets in the genesis of the larger oligomers. It's essential to discern whether the N-sheets play a proactive role in oligomerization or if they are reduced in the process. To shed light on this, MD simulations integrating A β 42 tetramers with both N-sheets and C-sheets within detergent micelles should be executed.
2. Prior investigations have indicated that A β 42 oligomers, when stabilized in SDS micelles, vary from dimers to tetramers. In contrast, those in DPC micelles are predominantly tetramers. Given the manuscript's assertion that A β 42 oligomerizes similarly in both SDS and DPC, an explanation addressing this divergence would be enlightening.
3. Upon extracting SDS via dialysis, 150-kD A β 42 oligomers are formed. This spurs the question: Have the authors examined the dimensions of A β 42 oligomers post DPC removal? Do the A β 42 oligomers consistently amass into 150kD structures, or might they maintain their configuration but diverge in size? Moreover, the authors' perspective on the aggregates' size ceiling would be of interest.
4. In Figure 2A, the gray shades are somewhat indistinct, overshadowed by the dominant red outlines. A refinement of this illustration for improved clarity would be beneficial.
5. The manuscript contains some typographic oversights, such as an unclear descriptor in the legend of Figure 6.

Reviewer #2 (Remarks to the Author):

This study by Muhammedkutty et al. sets out to investigate the mechanism by which SDS and DPC induce the formation of a stable, 150 kDa large Abeta oligomer, which has been studied extensively in the past. First, the authors use solid-state NMR to show that the oligomers made with the help of SDS and DPC are structurally indistinguishable (by NMR). Then they show, again using NMR, that beta-sheets that form in SDS are directly incorporated into the final, SDS free oligomer. These NMR data are complemented by MD simulations showing that only antiparallel beta-sheet tetramers are stable in SDS and DPC micelles whereas parallel tetramers or dimers are not.

This paper generally presents data giving very clean results such as the almost perfect overlap of NMR data from SDS and DPC oligomers or the stability of different beta-sheet starting structures in micelles

via MD. I support its publication if a few major concerns have been addressed:

Major comments

Results of the isotope mixing experiment very much depend on the definition of the error bar which is not given in the manuscript. Because the differences between pre and post SDS mixing are not huge, it will be important to know what the error bar is. Was this experiment done on a single biological replicate? In addition, it would be good to have an alternative parameter of statistical significance other than the visual inspection of the signal decay curves.

Another major concern is some over interpretation of the data in the discussion section. I wouldn't say that the authors "characterize(d) the detergent-assisted oligomerization pathway of A β 42". There are still too many steps in the process that have not been characterized. For example, the simulations only show that antiparallel tetramers are stable in micelles not that they actually form in them. Especially, because the known tetramer structure from DPC micelles is different from the tetramers used in the simulation. Similarly, the isotope mixing experiments don't show that C-sheet tetramers are directly incorporated into 150kDa oligomers. This interpretation is only compatible with the data.

Lastly, I think the story of the glycine holes is relatively mute. What is their biological significance? They are presented as remarkable but wouldn't it be more remarkable if the holes weren't filled by detergent? In the Discussion section the authors speculate that the absence of these glycine holes would allow parallel beta-sheets. Why not show this then with one more simulation?

Minor comments

Starting the results section with the the full interpretation of the data that follow is a bit strange. Doesn't this shut down any additional explanation for the data presented? Alternatively, if the authors want to start the paper with such as specific hypothesis, wouldn't they have to design experiments specifically to disprove their hypothesis?

Page 5: A few more words about how the samples were made would be helpful to understand this section.

Page 6: Mixing experiments are very hard to understand as described here. "...labeled and unlabeled monomers were mixed before introducing into SDS micelles..." introducing what?

Page 11: "Taken together, our MD simulations show that the antiparallel C-sheet tetramer is the only A β 42 small oligomer that is stable in SDS and DPC micelles." To make this statement, all possible small oligomers would have to be tested, which might be impossible.

Figure S4 H) is described twice and I) is not described.

We thank the reviewers for their constructive comments, which have helped improve the clarity of the experimental data and the overall presentation. Our point-by-point response is given below in blue.

Reviewer #1 (Remarks to the Author):

In the study by Zhou and colleagues, the prevalent pathways of A β 42 oligomerization in the presence of the detergents, SDS and DPC, were explored. The authors observed that A β 42 monomers cluster into tetramers that consist of antiparallel C-sheets. These tetramers gain stability from the interactions between GLY 33, 37, 38 and the detergent's hydrocarbon tails. With the support of SDS/DPC micelles, these A β 42 tetramers are fortified, and the elimination of SDS subsequently results in the emergence of 150-kD A β 42 oligomers. These insights are valuable for understanding A β oligomer aggregation in cellular environments and the formation of oligomers or fibrils influenced by specific sequences or residues. The paper is clearly presented, and the findings are systematically outlined. To enhance the quality of the work, I recommend addressing the following points:

1. It's worth noting that the tetramers, stabilized by either SDS or DPC and consisting of antiparallel C-sheets, could transition directly into larger oligomers. The tetramer structure highlighted in the study by Ciudad et al. showcases four antiparallel C-sheets alongside two N-sheets. This raises an intriguing query about the role of the N-sheets in the genesis of the larger oligomers. It's essential to discern whether the N-sheets play a proactive role in oligomerization or if they are reduced in the process. To shed light on this, MD simulations integrating A β 42 tetramers with both N-sheets and C-sheets within detergent micelles should be executed.

We did not perform MD simulations of tetramers with both N- and C-sheets within detergent micelles for the following reasons. First off, in their NMR structure, Ciudad et al. (ref. 20) already ruled out N-sheets inside DPC micelles. More importantly, the detergent micelles are simply too small to accommodate both the N-sheets and C-sheets inside.

2. Prior investigations have indicated that A β 42 oligomers, when stabilized in SDS micelles, vary from dimers to tetramers. In contrast, those in DPC micelles are predominantly tetramers. Given the manuscript's assertion that A β 42 oligomerizes similarly in both SDS and DPC, an explanation addressing this divergence would be enlightening.

Prior studies in SDS did not have the resolution to nail down the exact size and could only provide a range. The Ciudad study in DPC was able to identify tetramer as the dominant species.

3. Upon extracting SDS via dialysis, 150-kD A β 42 oligomers are formed. This spurs the question: Have the authors examined the dimensions of A β 42 oligomers post DPC removal? Do the A β 42 oligomers consistently amass into 150kD structures, or might they maintain their configuration but diverge in size? Moreover, the authors' perspective on the aggregates' size ceiling would be of interest.

Yes, we have confirmed that the A β 42 oligomers from both SDS and DPC preparations have similar sizes (text added in p. 5). We now also add our speculation as to why there exists a size

ceiling in these A β 42 oligomers (p. 15, first paragraph). In our idealized structural model (illustrated schematically in Figure 1), the 150-kD oligomer comprises monomers with two distinct conformations (“L” shape and “U” shape), in perfect order. In reality, monomer addition may not follow this perfect order. Such defects are favored from entropic consideration but reduce oligomer stability. The balance between entropy and stability determines the oligomer size.

4. In Figure 2A, the gray shades are somewhat indistinct, overshadowed by the dominant red outlines. A refinement of this illustration for improved clarity would be beneficial.

The gray contours are somewhat indistinct because they match too well with the red contours! We have now darkened the gray contours to black and added Supplementary Figure 2, which presents the two spectra side by side instead of overlaid, clearly indicating their similarity.

5. The manuscript contains some typographic oversights, such as an unclear descriptor in the legend of Figure 6.

We have corrected this and a few other typos.

Reviewer #2 (Remarks to the Author):

This study by Muhammedkutty et al. sets out to investigate the mechanism by which SDS and DPC induce the formation of a stable, 150 kDa large Abeta oligomer, which has been studied extensively in the past. First, the authors use solid-state NMR to show that the oligomers made with the help of SDS and DPC are structurally indistinguishable (by NMR). Then they show, again using NMR, that beta-sheets that form in SDS are directly incorporated into the final, SDS free oligomer. These NMR data are complemented by MD simulations showing that only antiparallel beta-sheet tetramers are stable in SDS and DPC micelles whereas parallel tetramers or dimers are not.

This paper generally presents data giving very clean results such as the almost perfect overlap of NMR data from SDS and DPC oligomers or the stability of different beta-sheet starting structures in micelles via MD. I support its publication if a few major concerns have been addressed:

Major comments

Results of the isotope mixing experiment very much depend on the definition of the error bar which is not given in the manuscript. Because the differences between pre and post SDS mixing are not huge, it will be important to know what the error bar is. Was this experiment done on a single biological replicate? In addition, it would be good to have an alternative parameter of statistical significance other than the visual inspection of the signal decay curves.

We now indicate in the Figure 3 caption that the error bars represent 95% confidence intervals in a nonlinear fit of NMR peaks. As for replication, we have published PITHIRDS-CT decay

curves for 100%-labeled V36 $^{13}\text{C}\alpha$ and A30 $^{13}\text{C}\beta$ in a 2013 JMB paper (ref. 18). These were measured on samples different from the one reported here but the results are very similar to those in Figure 3. Moreover, in the present study, we took great pains to minimize sample-to-sample variations for the three samples that we report in Figure 3. The close superposition of the three decays curves in Figure 3c demonstrate that we have achieved this goal (bottom of p. 8 to top of p. 9). To define a parameter for quantifying the difference in decay between the pre- and post-SDS samples, we now fit the decay curves to a Lorentzian function and report the resulting decay times, which do show significant differences (30 ± 1 ms vs. 36 ± 1 ms; p. 8, second paragraph).

Another major concern is some over interpretation of the data in the discussion section. I wouldn't say that the authors "characterize(d) the detergent-assisted oligomerization pathway of A β 42". There are still too many steps in the process that have not been characterized. For example, the simulations only show that antiparallel tetramers are stable in micelles not that they actually form in them. Especially, because the known tetramer structure from DPC micelles is different from the tetramers used in the simulation. Similarly, the isotope mixing experiments don't show that C-sheet tetramers are directly incorporated into 150kDa oligomers. This interpretation is only compatible with the data.

We have now changed "characterize" to "investigate".

Lastly, I think the story of the glycine holes is relatively mute. What is their biological significance? They are presented as remarkable but wouldn't it be more remarkable if the holes weren't filled by detergent? In the Discussion section the authors speculate that the absence of these glycine holes would allow parallel beta-sheets. Why not show this then with one more simulation?

We have now performed MD simulations for parallel C-sheets of A β 40 and A β 42 with Gly37 and Gly38 mutated to Val. The preliminary results (new Supplementary Fig. 11) support our speculation that, with the Gly-to-Val mutations, the parallel C-sheets could be stable in SDS micelles (bottom of p. 15 to top of p. 16).

Minor comments

Starting the results section with the the full interpretation of the data that follow is a bit strange. Doesn't this shut down any additional explanation for the data presented? Alternatively, if the authors want to start the paper with such as specific hypothesis, wouldn't they have to design experiments specifically to disprove their hypothesis?

Indeed this style of presentation is not common. We chose it because there are so many pieces to the puzzle and we wanted to provide readers with a map to make it easier for them to follow. We have designed many experiments to demonstrate that our conclusion is logical. In particular, our PITHIRDS-CT experiment disproves the scenario where detergent-stabilized tetramers disassemble before reassembling into the 150-kD oligomer. Our MD simulations disapprove the scenario where N-sheets are stabilized by detergent micelles.

Page 5: A few more words about how the samples were made would be helpful to understand this section.

We have now added a paragraph on the samples (p. 5).

Page 6: Mixing experiments are very hard to understand as described here. "...labeled and unlabeled monomers were mixed before introducing into SDS micelles..." introducing what?

We have rephrased this part to increase clarity.

Page 11: "Taken together, our MD simulations show that the antiparallel C-sheet tetramer is the only A β 42 small oligomer that is stable in SDS and DPC micelles." To make this statement, all possible small oligomers would have to be tested, which might be impossible.

We have now weakened the statement to: "... is likely the only ..."

Figure S4 H) is described twice and I) is not described.

We have now corrected this typo.

REVIEWERS' COMMENTS:

Reviewer #1 (Remarks to the Author):

I don't have further question. I support publication of this work in Communications Biology.

Reviewer #2 (Remarks to the Author):

The authors addressed all issues brought up in my original review. I recommend the publication of this article.